# GC-MIXER: A NOVEL ARCHITECTURE FOR TIME-VARYING GRANGER CAUSALITY INFERENCE

## ABSTRACT

The neural network has emerged as a practical approach to evaluate the Granger causality in multivariate time series. However, most existing studies on Granger causality inference are based on time-invariance. In this paper, we propose a novel MLP architecture, Granger Causality Mixer (GC-Mixer), which extracts parameters from the ~~weight matrix~~ causal matrix and imposes the hierarchical group lasso penalty on these parameters to infer time-invariant Granger causality and automatically select time lags. Furthermore, we extend GC-Mixer by introducing a multi-level fine-tuning algorithm to split time series automatically and infer time-varying Granger causality. We conduct experiments on the VAR and Lorenz-96 datasets, and the results show that GC-Mixer achieves outstanding performances in Granger causality inference.

## 1 INTRODUCTION

Granger causality is a statistical framework for analyzing the causality between time series. It offers a powerful tool to investigate temporal dependencies and infer the directionality of influence between variables (Maziarz, 2015; Friston et al., 2014; Shojaie & Fox, 2022). By examining the past values of a series, Granger causality seeks to determine if the historical knowledge of one variable improves the prediction of another (Bressler & Seth, 2011; Barnett & Seth, 2014). Revealing inner interactions from observational time series has made Granger causality useful for the investigation in many fields, such as econometrics (Mele et al., 2022), neuroscience (Chen et al., 2023b), climate science (Ren et al., 2023), etc.

The Granger causality inference has conventionally relied on linear methods, such as the Vector Autoregressive (VAR) (Seth et al., 2015; Rossi & Wang, 2019). However, due to the prevalence of nonlinearity in most time series, applying linear methods to analyze nonlinear time series may lead to false Granger causality inference. Consequently, there has been a growing interest in incorporating the neural network into the study of Granger causality, owing to the inherent nonlinear mapping capability (Marcinkevičs & Vogt, 2021a). Recently, the Multi-Layer Perceptron (MLP) and Long Short-Term Memory (LSTM) have emerged as prominent choices, garnering significant attention in related research. Tank et al. (2021) propose component-wise MLP (cMLP) and LSTM (cLSTM), which extract Granger causality from the first layer weights in the neural network and impose the sparse penalties on weights to infer Granger causality. Nauta et al. (2019) proposes the Temporal Causal Discovery Framework (TCDF) based on the convolutional neural network and attention mechanism, which automatically infer the time lag by looking up the highest kernel weight of the input time series.

Although the models mentioned above can effectively infer Granger causality in time series, there are still some limitations. Granger causality is time-varying in the real-world scenario, (Lu et al., 2014; Li et al., 2018), whereas these models assume the Granger causality is time-invariant. In addition, even if the time series or its inner causal relationships change slightly, these models still need to reselect appropriate hyperparameters. Otherwise, the inference accuracy will fluctuate wildly. For these models, inferring time-varying Granger causality requires constantly changing hyperparameters, which is impractical in the real-world scenario. In this paper, we propose GC-Mixer for Granger causality inference. We modify the configurations of the time series, and GC-Mixer can maintain stable performance without changing the hyperparameters. Furthermore, we extend the

model for the time-varying scenario by introducing a multi-level fine-tuning algorithm. Our main contributions can be summarized as follows:

- We propose GC-Mixer, a novel model for time-invariant Granger causality inference. ~~The model is less susceptible to the influence of the group lasso hyperparameter, making it well-suited for Granger causality inference.~~ Our model applies a new approach to extract Granger causality from the output of the Mixer Block, which is different from existing models.

- ~~A multi-level fine-tuning algorithm is proposed to extend GC-Mixer for automatic splitting time series to infer time-varying Granger causality.~~ A multi-level fine-tuning algorithm is proposed as an extension of GC-Mixer for automatic splitting time series to infer time-varying Granger causality, which solves the problem that the optimal number of split sequences is difficult to determine in the traditional manual splitting method.

- Experiments on VAR and Lorenz-96 datasets (Tank et al., 2021) validate that GC-Mixer attains stable and outstanding performances in time-invariant and time-varying Granger causality inference.

## 2 PRELIMINARY

### 2.1 VECTOR AUTOREGRESSIVE (VAR)

Assume a $p$-dimensional stationary time series $x_t$ with $T$ observation time point $(x_1, \ldots, x_t)$. In the VAR model, the $t^{th}$ time point $x_t$ can be written as a linear combination of the past $K$ lags of the time series:

$$x_t = \sum_{k=1}^{K} A^{(k)} x_{t-k} + e^t \tag{1}$$

where $A^{(k)} \in \mathbb{R}^{p \times p}$ is the regression coefficient matrix representing how time lag $k$ effects the future prediction, $e^t$ is zero means. To infer Granger causality in the VAR model, group lasso penalty is applied:

$$\min_{A^{(1)}, \ldots, A^{(K)}} \sum_{t=K}^{T} \|x_t - \sum_{k=1}^{K} A^{(k)} x_{t-k}\|_2^2 + \lambda \sum_{ij} \|A_{i,j}^{(1)}, \ldots, A_{i,j}^{(K)}\|_2 \tag{2}$$

where $\lambda$ is the hyperparameter that controls the level of penalty, $\| \cdot \|_2$ denoted as the $L2$ norm. In this model, if there exists a time lag $k$ for which $A_{i,j}^{(k)} \neq 0$, then time series $j$ Granger-cause to time series $i$.

### 2.2 NONLINEAR-AUTOREGRESSIVE (NAR)

Assume a $p$-dimensional non-stationary time series $x_t = [x_{<t1}, \ldots, x_{<tp}]$, where $x_{<ti} = (\ldots, x_{<(t-2)i}, x_{<(t-1)i})$. In the NAR model, the $t^{th}$ time point $x_t$ can be denoted as a function $g$ of its past time values:

$$x_t = g(x_{<t1}, \ldots, x_{<tp}) + e^t \tag{3}$$

The function $g$ takes the form of the neural network, such as MLP or LSTM. Similar to the VAR model, the inference of Granger causality in NAR can be denoted as:

$$\min_W \sum_{t=K}^{T} (x_t - g(x_{<t1}, \ldots, x_{<tp}))^2 + \lambda \sum_{j=1}^{p} \Psi(W_{:,j}) \tag{4}$$

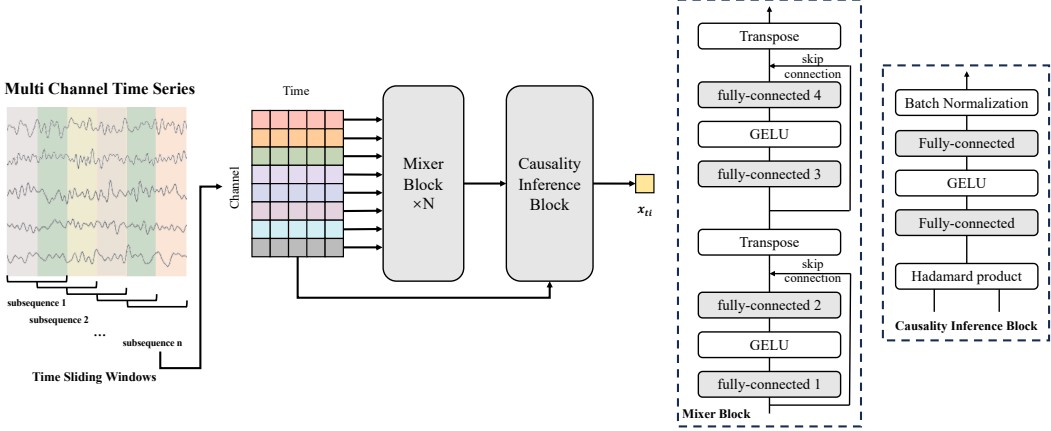

Figure 1: The architecture of GC-Mixer.

where $W$ is the weight matrix extracted from the neural network, $\Psi$ is the group lasso penalty that penalizes the parameters in $W$ to zero. In the NAR model, if there exists a time lag $k$, $W_{:,j}^{k}$ contains non-zero parameters, time series $j$ Granger-causes to time series $i$.

## 2.3 COMPONENT-WISE NAR

In the NAR model, it is assumed that the prediction of $x_{ti}$ depends on the same past time lag of all the series. Nevertheless, $x_{ti}$ may depend on different past-time lags from all series. To infer the Granger causality from different time series and time lags, $x_{ti}$ can be denoted by a nonlinear function $g_i$ as:

$$x_{ti} = g_i\left(x_{<t1}, \ldots, x_{<tp}\right) + e^{ti} \tag{5}$$

The Granger causality inference in component-wise NAR model is turned to:

$$\min_{W} \sum_{t=K}^{T} \left(x_{ti} - g_i\left(x_{<t1}, \ldots, x_{<tp}\right)\right)^2 + \lambda \sum_{j=1}^{p} \Psi\left(W_{:,j}\right) \tag{6}$$

## 3 PROPOSED METHOD

### 3.1 GC-MIXER ARCHITECTURE

The architecture of GC-Mixer is illustrated in Figure 1. It contains a time-sliding window, N stacks of Mixer Block, and a Causality Inference Block. Overlapped time subsequences obtained by the time-sliding window are respectively input into the model. They go through a stack of Mixer Block to fuse time and channel features, respectively. In the Causality Inference Block, the output of the Mixer Block computes the Hadamard product with the input time subsequence. The result is fed into an MLP with two fully-connected layers, a GELU activate function, and a batch normalization for predicting $x_{ti}$. ~~We define the output of the Mixer Block as the weight matrix corresponding to the input subsequence, which serves as the basis for Granger causality inference. Same as cMLP, the weight matrix is imposed on the hierarchical group lasso penalty for automatic time lag selection.~~ The components of GC-Mixer are shown as follows:

### 3.1.1 TIME-SLIDING WINDOWS

Assume a $p$-dimensional multivariate time series $x_t \in \mathbb{R}^{p \times T}$ with $T$ time samples in each dimension. The time window width is $K$, which equals the maximum time lag. The time step is one. As the time window slides through the time series, it generates $T - K + 1$ subsequences. These subsequences are denoted as $x^{(1)}, \ldots, x^{(T-K+1)} \in \mathbb{R}^{p \times K}$, respectively.

### 3.1.2 MIXER BLOCK

The Mixer Block stacks $N$ times. Each block includes four fully-connected layers, two GELU activation functions, and two residual connections. The block firstly projects $x^{(n)}$ along the time domain:

$$U^{(n)} = x^{(n)} + W_2 \rho \left( W_1 \left( x^{(n)} \right) \right) \tag{7}$$

where $\rho$ is the GELU activation function. Then the Mixer Block projects $U_t^{\hat{(n)}}$ along the channel domain:

$$Y^{(n)} = \hat{U^{(n)}} + W_4 \rho \left( W_3 \left( \hat{U^{(n)}} \right) \right) \tag{8}$$

where $\hat{U^{(n)}} \in \mathbb{R}^{K \times p}$ is the transpose of $U_t^{(n)}$. $Y^{(n)} \in \mathbb{R}^{K \times p}$ is transposed to have the same dimensions as the input subsequence $x_t^{(n)}$, which is denoted as ~~the weight matrix~~ $W^{(n)}$:

$$W^{(n)} = \hat{Y^{(n)}} \in \mathbb{R}^{p \times K} \tag{9}$$

### 3.1.3 CAUSALITY INFERENCE BLOCK

The Causality Inference Block includes two inputs: the subsequence $x^{(n)} \in \mathbb{R}^{p \times K}$ and ~~the weight matrix~~ $W^{(n)} \in \mathbb{R}^{p \times K}$. The Hadamard product of two matrices is computed, and the result is unfolded into a vector $M = \left( W_{11}^{(n)} x_{11}^{(n)}, \ldots, W_{pK}^{(n)} x_{pK}^{(n)} \right)$. Following the Equation 5, each component $x_i^{(n)}$ corresponding to a separate $g_i$. $g_i$ takes the form of an MLP with two fully-connected layers, a GULE ~~GULE~~ activation function, and batch normalization. Finally, $M$ is projected to the predicted $x_i^{(n)}$ through the $g_i$:

$$x_i^{(n)} = g_i \left( W_{i,(11)}^{(n)} x_{11}^{(n)}, \ldots, W_{i,(pK)}^{(n)} x_{pK}^{(n)} \right) + e^i \tag{10}$$

where $W_{i,(jk)}^{(n)}$ is denoted as the the $j$ row and $k$ column of $W$ corresponding to $g_i$ and time subsequence $x^n$. According to Equation 6, the inference of Granger causality in Equation 10 uses component-wise NAR combined with lasso penalty:

$$\min_W \sum_{n=1}^{T-K+1} \left( x_i^{(n)} - g_i \left( W_{i,(11)}^{(n)} x_{11}^{(n)}, \ldots, W_{i,(pK)}^{(n)} x_{pK}^{(n)} \right) \right)^2 + \lambda \sum_{j=1}^{p} \| W_{i,(j,:)}^{(n)} \|_F \tag{11}$$

where $\| \cdot \|_F$ is denoted as the Frobenius matrix norm. Meanwhile, a variant of group lasso called hierarchical group lasso is applied on GC-Mixer, which has a nested group structure and imposes a larger penalty for higher lags. The loss function is defined as:

$$\begin{aligned} \mathcal{L} = & \sum_{n=1}^{T-K+1} \left( x_i^{(n)} - g_i \left( W_{i,(11)}^{(n)} x_{11}^{(n)}, \ldots, W_{i,(pK)}^{(n)} x_{pK}^{(n)} \right) \right)^2 \\ & + \sum_{n=1}^{T-K+1} \sum_{j=1}^{p} \sum_{k=1}^{K} \lambda \| W_{i,(j,k)}^{(n)}, \ldots, W_{i,(j,K)}^{(n)} \|_F \end{aligned} \tag{12}$$

We define $W$ as the causal matrix, which serves as the basis for Granger causality inference. In our practice, the sparse penalty cannot penalize the parameters in the $W$ to zero~~, which is the same in cMLP and cLSTM~~. Therefore, if and only if for all subsequences $n$ and lag $K$, the $F$-norm of $W_{i,(j,k)}^{(n)}$ more than a threshold $\epsilon$, series $j$ Granger-causes series $i$:

$$\sum_{n}^{T-K+1} \sum_{k=1}^{K} \| W_{i,(j,k)}^{(n)} \|_F \geq \epsilon \tag{13}$$

We respectively impose the group lasso and hierarchical group lasso on GC-Mixer and cMLP ~~and cLSTM~~ and find that all models perform better under the hierarchical group lasso. Therefore, the following sections uniformly use the hierarchical group lasso as the sparse penalty.

---

**Algorithm 1** Multi-level fine-tuning algorithm

---

1: $\mathcal{L}_{best} \leftarrow \infty$
2: $i \leftarrow 1$
3: Pre-training $x_t$ according to Equation 12 to get the loss $\mathcal{L}$.
4: **while** $\mathcal{L} < \mathcal{L}_{best}$ **do**
5:    $L_{best} = L$
6:    $i = i + 1$
7:    Separate input time series $x_t$ into $2^{i-1}$ target time series.
8:    Fine-tuning each target time series according to Equation 12 to get the corresponding loss $\mathcal{L}_1, \mathcal{L}_2, \ldots, \mathcal{L}_{2^{i-1}}$.
9:    $\mathcal{L} = (\mathcal{L}_1 + \mathcal{L}_2 +, \ldots, + \mathcal{L}_{2^{i-1}})/2^{i-1}$
10: **end while**
11: Inferring Granger causality of each target time series according to Equation 13.
12: **return** Time-varying Granger causality inference.

---

## 3.2 EXTEND GC-MIXER FOR TIME-VARYING GRANGER CAUSALITY INFERENCE

An existing approach to infer time-varying Granger causality is to separate the input time series into multiple segments and input these segments into the neural network to obtain a series of Granger causality interactions(Shan et al., 2023; Ren et al., 2020). However, this method requires manually separating time series. In this section, we propose a multi-level fine-tuning algorithm to automatically separate time series and extend our GC-Mixer for more accurate time-varying Granger causality inference, as shown in Figure 2.

In the pre-training stage, the input time series $x_t$ is trained on GC-Mixer according to Equation 12 to capture its general features of the time series. In the multi-level fine-tuning stage, for each level $i(i \geq 2)$, we build $2^{i-1}$ target GC-Mixer models and separate the input time series $x_t$ into $2^{i-1}$ target time series. All the weights and biases learned from the previous level are transferred to the target models in the current level for further training. We evaluate whether the averaged fine-tuned loss of each level is less than the previous level, and if so, we continue to separate the time series until the loss is no longer reduced. Finally, the Granger causality corresponding to each target time series is inferred using Equation 13. By pooling these results together, we obtain the time-varying Granger causality inference. The time complexity of the algorithm is $O\left((T - K + 1) \times p \times (2^i - 1)\right)$. The detailed algorithm is demonstrated in Algorithm 1.

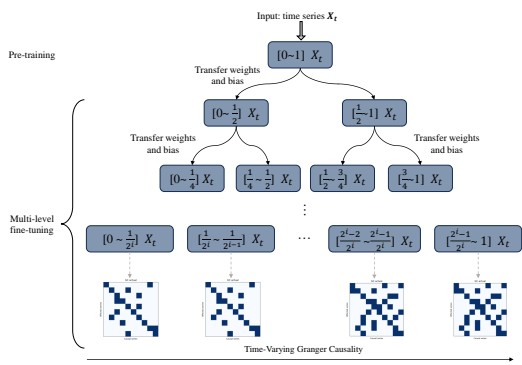

Figure 2: Multi-level fine-tuning.

We fine-tune the entire neural network rather than only the final output layer. Through multi-level fine-tuning, the input time series can be automatically separated into multiple target time series, allowing for more accurate time-varying Granger causality inference.

## 4 EXPERIMENTS

### 4.1 DATASETS

The first dataset is the VAR. For a $p$-dimensional time series $x_t$, the vector autoregressive model is given by:

$$x_t = A^{(1)}x_{t-1} + A^{(2)}x_{t-2} +, \ldots, + A^{(k)}x_{t-k} + u_t \tag{14}$$

Table 1: VAR (3), $T = 1000$, $p = 10$

| MODEL | AUROC | | | |
|---|---|---|---|---|
| | $sparsity = 0.2$ | $sparsity = 0.3$ | $sparsity = 0.4$ | $sparsity = 0.5$ |
| cMLP | **1** | 0.62 | 0.60 | 0.58 |
| cLSTM | 0.91 | 0.59 | 0.57 | 0.56 |
| TCDF | 0.85 | 0.53 | 0.51 | 0.47 |
| GVAR | 1 | 1 | 0.98 | 0.95 |
| GC-Mixer | **1** | **1** | **1** | **0.99** |

where $(A^{(1)}, A^{(2)}, \ldots, A^{(k)})$ are regression coefficients matrices and $u_t$ is a vector of errors with Gaussian distribution. We define $sparsity$ as the percentage of non-zero coefficients in $A^{(i)}$, and different $sparsity$ represent different quantities of Granger causality interaction in the VAR model. The second dataset is the Lorenz-96, which is a mathematical model used to study the dynamics of a simplified atmospheric system. For $p$-dimensional Lorenz-96 model, its ordinary differential equation is given by:

$$\frac{\partial x_{t,i}}{\partial t} = -x_{t,i-1}\left(x_{t,i-2} - x_{t,i+1}\right) - x_{t,i} + F \tag{15}$$

where $F$ represents the forcing term applied to the system. The values of $p$ and $F$ impact the behavior of the Lorenz-96 model. Increasing $F$ makes the system more chaotic, while changing $p$ affects the spatial complexity of the system.

## 4.2 Model Evaluation

We compare the proposed GC-Mixer with cMLP, cLSTM, GVAR, and TCDF. Our goal is to compare the ability of models to maintain stable Granger causality inference accuracy with unchanged hyperparameters. We search for the best-performing hyperparameters for each model in VAR (3) with the $sparsity$ of 0.2 and Lorenz-96 with the force term $F$ of 10. Subsequently, with the hyperparameters unchanged, the configurations of the VAR and Lorenz-96 datasets are changed under the following three conditions:

1. To simulate different Granger causality interaction quantities in time series, the $sparsity$ of the regression coefficient matrix in VAR (3) is increased from 0.2 to 0.3, 0.4, and 0.5 while keeping the dimension $p$ fixed at 10.

2. To test the model's performances under different channel dimensions, the dimension $p$ of VAR (3) is modified from 10 to 15, 20, and 25 while maintaining the $sparsity = 0.2$.

3. To simulate the different strengths of nonlinearity in the causal interactions between the variables, the forcing term $F$ of Lorenz-96 is adjusted from 10 to 20, 30, and 40 while the dimension $p$ remains 10.

We use the True Positive Rate (TPR) and False Positive Rate (FPR) according to AUROC (Area Under the Receiver Operating Characteristic Curve). The AUROC is generated with one $\lambda$ value and sweep threshold $\epsilon$. The results on the VAR dataset are presented in Table 1, Table 2. The performances of four models are close when $sparsity = 0.2$. As time series have more Granger causality interactions ($sparsity = 0.5$), the AUROC scores of cMLP, cLSTM, and TCDF decrease significantly. In contrast, GC-Mixer and GVAR maintain a stable performance, with AUROC only reducing from 1 to 0.99 and 1 to 0.95. A similar observation arises when dimension $p$ increases to 25. GC-Mixer maintains a high AUROC of 0.96, and GVAR achieves an AUROC of 0.93, while cMLP, cLSTM, and TCDF cannot infer Granger causality effectively, with AUROC of 0.47, 0.49, and 0.48.

The results on the Lorenz-96 dataset are shown in Table 3. In the case of $F = 10$, GVAR achieves the highest AUROC of 0.99, and GC-Mixer achieves an AUROC score of 0.94, while the AUROC of cMLP and cLSTM are 0.96 and 0.94, respectively. However, when $F = 30$, both GC-Mixer and GVAR have a significant decrease in AUROC, while cMLP and cLSTM still have a stable

Table 2: VAR (3), $T = 1000$, $sparsity = 0.2$

| MODEL | AUROC | | | |
|---|---|---|---|---|
| | $p = 10$ | $p = 15$ | $p = 20$ | $p = 25$ |
| cMLP | 1 | 0.49 | 0.49 | 0.47 |
| cLSTM | 0.91 | 0.54 | 0.49 | 0.49 |
| TCDF | 0.85 | 0.53 | 0.51 | 0.48 |
| GVAR | 1 | 0.99 | 0.98 | 0.93 |
| GC-Mixer | 1 | 1 | **0.99** | **0.96** |

Table 3: Lorenz-96, $T = 1000$, $p = 10$

| MODEL | AUROC | | | |
|---|---|---|---|---|
| | $F = 10$ | $F = 20$ | $F = 30$ | $F = 40$ |
| cMLP | 0.96 | 0.95 | **0.94** | **0.95** |
| cLSTM | 0.94 | 0.93 | 0.92 | 0.90 |
| TCDF | 0.73 | 0.71 | 0.69 | 0.68 |
| GVAR | 0.99 | 0.96 | 0.85 | 0.78 |
| GC-Mixer | 0.94 | 0.92 | 0.80 | 0.73 |

performance. When $F = 40$, the performance of our model still lags behind cMLP cLSTM and GVAR, which may be attributed to the fact that GC-Mixer has more parameters than cMLP and cLSTM, leading to more prone to overfit.

## 4.3 AUTOMATIC LAG SELECTION

We compare GC-Mixer with cMLP on automatic lag selection using the hierarchical group lasso penalty. It is important to note that we do not consider cLSTM, TCDF models, and Lorenz-96 dataset in this experiment since cLSTM cannot output time lag, TCDF can output only one lag of a time series, and Lorenz-96 does not have time lag according to Equation 15.

As in the previous section, we conduct experiments on VAR (3) with $sparsity = 0.2$ and $sparsity = 0.3$. The maximum lag order of $K$ is 5. The time lag selection of each model in 10 channels and the corresponding true results are shown in Figure 3 and Figure 4.

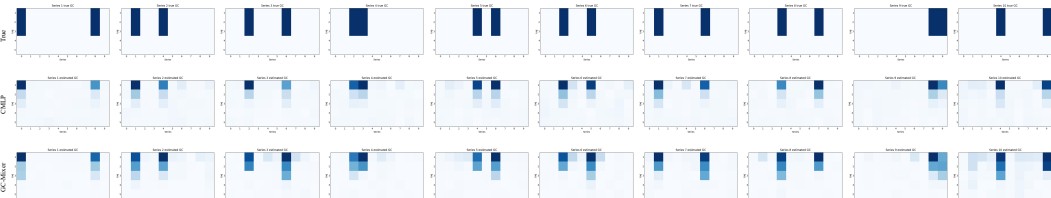

Figure 3: (Top) The true results of ten output series. (Middle) Ten output series results of cMLP inference. (Bottom) Ten output series results of GC-Mixer inference. The comparison of qualitative results between GC-Mixer with cMLP in the automatic lag selection. The rows of each image correspond to the time lag, with $k = 1$ on the top and $k = 5$ on the bottom, while the columns correspond to 10 different input series, with series one on the left and series ten on the right. The results are computed by the $L2$ norm of the weights in the neural network. The brighter color means that the corresponding time lag has a more significant impact on future prediction.

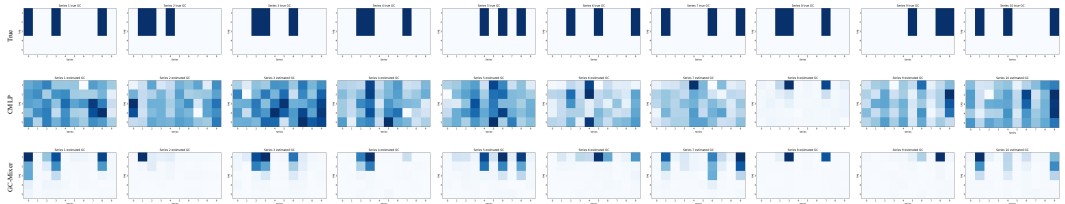

Figure 4: Automatic lag selection on VAR (3) with the $sparsity = 0.3$. (Top) The true results of 10 output series. (Middle) Ten output series results of cMLP inference. (Bottom) Ten output series results of GC-Mixer inference.

The results show that GC-Mixer performs better, selecting appropriate time lags for most channels in the time series. In the case of VAR (3) with the $sparsity = 0.2$, both GC-Mixer and cMLP can correctly select most of the time lags. GC-Mixer performs even better on the higher lag selection, especially on channels 2, 7, 8, and 10. In the case of VAR (3) with the $sparsity = 0.3$, cMLP can only effectively select time lag in channel 8, while GC-Mixer accurately selects most of the time lags, though there exist a few false time lag selections in channels 2, 6, 8. These false selections can primarily be attributed to the choice of hyperparameters of the hierarchical group lasso penalty. If the hyperparameter is excessively large, it penalizes too high on later time lag orders, resulting in false lag selection.

## 4.4 TIME-VARYING GRANGER CAUSALITY INFERENCE

We formulate four scenarios to evaluate the performances of the proposed multi-level fine-tuning algorithm on time-varying Granger causality inference. Each scenario consists of two time series with $T = 1000$, containing two different types of Granger causality:

1. The first scenario contains a preceding VAR (2) time series followed by a VAR (3) time series, and both of the $sparsity = 0.2$.

2. The second scenario involves a preceding time series generated by VAR (3) with the $sparsity = 0.2$, followed by VAR (3) with a $sparsity = 0.3$.

3. The third scenario contains a preceding Lorenz-96 time series with $F = 10$ followed by a Lorenz-96 time series with $F = 20$.

4. The fourth scenario includes a preceding VAR (3) time series with the $sparsity = 0.5$ followed by a Lorenz-96 time series with $F = 10$.

Existing machine learning-based Granger causality inference models like cMLP and cLSTM do not achieve time-varying Granger causality inference, resulting in these models not being compared with GC-Mixer directly. Therefore, for each scenario, we manually split the time series into two segments and apply cMLP, cLSTM, and GC-Mixer on these segments to achieve a simple time-varying Granger causality inference. Then, based on the multi-level fine-tuning algorithm, GC-Mixer automatically splits the whole time series and infers time-varying Granger causality. The results are illustrated in Figure 5, Table 4, and Figure 8 in Appendix D. The corresponding ROC curves of each scenario are shown in Figure 6 and Figure 7 in the Appendix C.

Table 4: AUROC of the four scenarios in time-varying Granger causality inference

| Model | Algorithm | Scenario 1 | Scenario 2 | Scenario 3 | Scenario 4 |
|---|---|---|---|---|---|
| | | AUROC | AUROC | AUROC | AUROC |
| cMLP | Manual splitting | 0.98 | 0.59 | 0.69 | 0.62 |
| cMLP | Multi-level fine-tuning (Automatic splitting) | **0.99** | 0.64 | 0.72 | 0.55 |
| cLSTM | Manual splitting | 0.67 | 0.54 | **0.99** | 0.63 |
| cLSTM | Multi-level fine-tuning (Automatic splitting) | 0.48 | 0.39 | 0.92 | **0.76** |
| GC-Mixer | Manual splitting | 0.98 | 0.95 | 0.89 | 0.63 |
| GC-Mixer | Multi-level fine-tuning (Automatic splitting) | **0.99** | **0.99** | 0.92 | 0.65 |

The results indicate that GC-Mixer performs better than cMLP and cLSTM in scenarios 1, 2, and 4 using the splitting time series manually. Specifically, When the time lags in the time series changes, the AUROC scores of GC-Mixer and cMLP are 31% higher than cLSTM. When the $sparsity$ of Granger causality in the time series is changed, the AUROC score of GC-Mixer is higher than those in cMLP and cLSTM with 36%, 41% increments. When the nonlinear strength of the time series changes, the performances of GC-Mixer and cLSTM are close, which are 20% and 30% higher than cMLP, respectively. For scenario 4, the AUROC of GC-Mixer also increases 2% than cMLP.

Moreover, we also conduct our algorithm on GC-Mixer, cMLP, and cLSTM. The proposed algorithm further improves the AUROC score for GC-Mixer with 1%, 4%, 4%, and 2% increments in four scenarios.For cMLP, the algorithm improves the AUROC score with 1%, 5%, and 3% increments in scenarios 1, 2, and 3. However, the algorithm only improves the AUROC score of cLSTM

in scenario 4 with 11% increments. Therefore, our algorithm can effectively extend GC-Mixer for inferring time-varying Granger causality when the time series contains different time lags, different strengths of nonlinearity, different quantities of causal interactions, and the linear-to-nonlinear transition scenario. For other models, our algorithm can improve performance in specific scenarios.

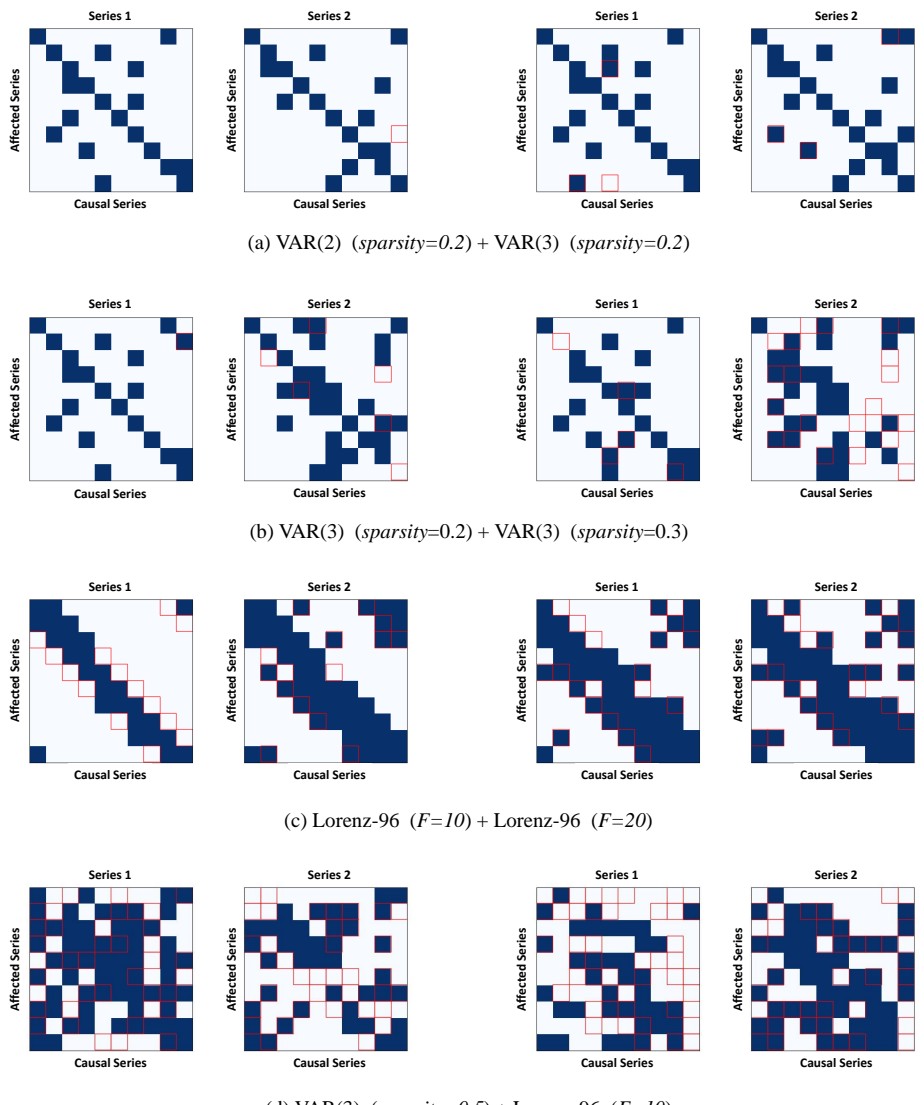

(a) VAR(2) (*sparsity=0.2*) + VAR(3) (*sparsity=0.2*)

(b) VAR(3) (*sparsity=0.2*) + VAR(3) (*sparsity=0.3*)

(c) Lorenz-96 (*F=10*) + Lorenz-96 (*F=20*)

(d) VAR(3) (*sparsity=0.5*) + Lorenz-96 (*F=10*)

Figure 5: Time-varying Granger causality inference. (Left) The two columns are inferred using the multi-level fine-tuning algorithm on GC-Mixer. (Right) The two columns are inferred by GC-Mixer using splitting time series manually. The blue blocks indicate that Granger causality relationship exists between two time series. The white blocks indicate no Granger causality relationship between two time series. The blocks surrounded by the red line are the false Granger causality inferences.

## 5  CONCLUSION

In this paper, we propose the Granger Causality Mixer (GC-Mixer), a novel framework for time-varying Granger causality inference, which applies an all-MLP architecture without using convolution and self-attention. The model maintains a stable performance without changing the group lasso hyperparameter, even if the quantities of Granger causality interaction, channel dimensions, or nonlinearity strengths in time series are changed. Using the hierarchical group lasso penalty, GC-Mixer

automatically selects time lags and achieves more accurate lag selections than existing models. To attain time-varying Granger causality inference, we propose a multi-level fine-tuning algorithm that exhibits outstanding performances on various conditions and enhances the capability of the model for the real-world scenario.

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

# A REVIEW OF RELATED WORKS

## A.1 NEURAL NETWORK-BASED GRANGER CAUSALITY

Neural network-based Granger causality has gained significant attention in the community due to its ability to infer Granger causality in time series. However, although many neural networks, such as MLP (Challu et al., 2023; Hamzaçebi et al., 2009), LSTM (Xu et al., 2020; Song et al., 2018), and Transformer (Zhou et al., 2021; Li et al., 2019), have shown excellent performances in time series forecasting, they remain black-box, making them difficult to explain the inner structure in multivariate time series. Thus, compared with the vector autoregressive, the major challenge encountered in these models is how to extract Granger causality from the neural network. To solve this problem, Tank et al. (2021) proposes two component-wise architecture called cMLP and cLSTM, which extracts Granger causality from the first layer weights of the neural network and imposes a sparsity-inducing penalty on weights to select time lag automatically. ~~The models experiment with three types of penalties, among which the hierarchical penalty demonstrates the best performance.~~ Fan et al. (2023) proposes MSNGC with a consistency threshold to infer binary Causality. The model uses different receptive fields to infer Granger causality and fuses them through learned attention weight for getting better performance. Yin & Barucca (2022) proposes a deep recurrent neural network incorporating a latent confounding module, which captures the influence of unobserved variables on the Granger causality between time series. Khanna & Tan (2019) proposes economy-SRU, which can directly infer Granger causality from the structured sparsity of SRU network parameters. The model sets group-sparse regularization combined with column sparsity on the weight coefficients. Marcinkevičs & Vogt (2021b) proposes GVAR to infer multivariate Granger causality from the nonlinear system. The model is based on an extension of self-explaining neural networks, leading to more interpretable than other neural network-based Granger causality inference models.

## A.2 TIME-VARYING GRANGER CAUSALITY

Recently, time-varying Granger causality has been widely studied. Vector autoregressive, lasso regression are two primary efforts. By introducing time-varying parameters into the VAR model, Li et al. (2012) proposes a parametric nonlinear Granger causality model and used regularized orthogonal least squares to reduce the number of parameters in the model. Sato et al. (2006) captures the continuous time-varying Granger causality by applying wavelet expansion. However, these methods infer the Granger causality only in the bivariate system. A practical solution for high-dimensional systems is applying lasso regression regularization. Gao & Yang (2022) proposes a kernel-reweighted group lasso to model time-varying Granger causality. Ren et al. (2020) proposes another group lasso algorithm, the Hilbert-Schmidt independence criterion lasso, to infer nonlinear Granger causality among multivariate time series. Nevertheless, these methods need to select the lag manually rather than automatically, which cannot satisfy many practical situations.

## A.3 FINE-TUNING IN THE DEEP NEURAL NETWORK.

Fine-tuning is a critical training method in transfer learning that involves pre-training the source task and specific its performance by further training it on the target task. Carneiro et al. (2015) proposes a fine-tuning CNN model that replaces the fully-connected layers on the pre-trained model with a logistic regression layer. Then the dataset is used to fine-tune only the appended layer, with the other layer of the network unchanged. Different from the many fine-tuning approaches which contain single level, Wu & Zhao (2022) proposes a Transformer-based multi-level fine-tuned model including pre-training on datasets to capture overall spatiotemporal features and multi-level fine-tuning to explore the intra-frame spatial information. Bayer et al. (2022) introduces a four-level fine-tuning method, where higher levels of fine-tuning result in the model becoming more specialized, leading to the actual task with less data.

Our motivation to study GC-Mixer arises from the need to infer time-varying Granger causality in multivariate time series. While similar architectures have been utilized for various scenarios, such as the well-known MLP Mixer architecture in computer vision (Yu et al., 2022), they have also been applied to time series forecasting (Chen et al., 2023a), NLP (Fusco et al., 2022), and image classification (Zhang et al., 2022). To our knowledge, utilizing an MLP Mixer-based architecture

specifically for inferring Granger causality in time series has not been previously explored in the existing literatures.

## B  HYPERPARAMETERS AND CONFIGURATIONS

Table 5 to 8 illustrate the configuration and hyperparameters of the different models used for comparison in Section 4. In the experiments, we found that the performances of cMLP and cLSTM are also strongly affected by the random seed used to generate the time series. By changing the random seed, the results of cMLP and cLSTM fluctuate wildly. Therefore, we uniformly use the $seed = 0$ in all experiments, which is provided in the code of Tank et al. (2021).

Table 5: The detailed parameters and configurations of the cMLP model used in this paper.

| Parameters | Dataset | | |
| --- | --- | --- | --- |
| | **Var (3)** $sparsity = 0.2/0.3/0.4/0.5$ | **VAR (3)** $p = 10/15/20/25$ | **Lorenz-96** $F = 10/20/30/40$ |
| Batch size | 1000 | 1000 | 1000 |
| Ridge regularization | - | - | 0.001 |
| Group lasso hyperparameter | 0.002 | 0.002 | 1 |
| Maximum time lag | 5 | 5 | - |
| Training epochs | 2000 | 2000 | 2000 |
| Learning rates | 0.01 | 0.01 | 0.01 |

Table 6: The detailed parameters and configurations of the cLSTM model used in this paper.

| Parameters | Dataset | | |
| --- | --- | --- | --- |
| | **Var (3)** $sparsity = 0.2/0.3/0.4/0.5$ | **VAR (3)** $p = 10/15/20/25$ | **Lorenz-96** $F = 10/20/30/40$ |
| Batch size | 1000 | 1000 | 1000 |
| Ridge regularization | - | - | 0.001 |
| Group lasso hyperparameter | 0.002 | 0.002 | 1 |
| Maximum time lag | - | - | - |
| Training epochs | 2000 | 2000 | 2000 |
| Learning rates | 0.01 | 0.01 | 0.01 |

Table 7: The detailed parameters and configurations of the TCDF model used in this paper.

| Parameters | Dataset | | |
| --- | --- | --- | --- |
| | **Var (3)** $sparsity = 0.2/0.3/0.4/0.5$ | **VAR (3)** $p = 10/15/20/25$ | **Lorenz-96** $F = 10/20/30/40$ |
| Kernel size | 2 | 2 | 4 |
| Batch size | 1000 | 1000 | 1000 |
| Layers | 3 | 3 | 2 |
| Dilation | 1 | 1 | 1 |
| Significance | 8 | 8 | 0.8 |
| Maximum time lag | 5 | 5 | - |
| Training epochs | 2000 | 2000 | 2000 |
| Learning rates | 0.01 | 0.01 | 0.01 |

Table 8: The detailed parameters and configurations of the GC-Mixer model used in this paper.

| Parameters | Dataset | | |
| --- | --- | --- | --- |
| | **Var (3)** $sparsity = 0.2/0.3/0.4/0.5$ | **VAR (3)** $p = 10/15/20/25$ | **Lorenz-96** $F = 10/20/30/40$ |
| Batch size | 1000 | 1000 | 1000 |
| Ridge regularization | - | - | - |
| Group lasso hyperparameter | 5e-07 | 5e-07 | 0.025 |
| Maximum time lag | 5 | 5 | - |
| Training epochs | 120 | 120 | 120 |
| Learning rates | 0.01 | 0.01 | 0.01 |

# C  ROC PLOTS

The comparisons of the ROC curve in Section 4.4 are shown in Figure 6, Figure 7.

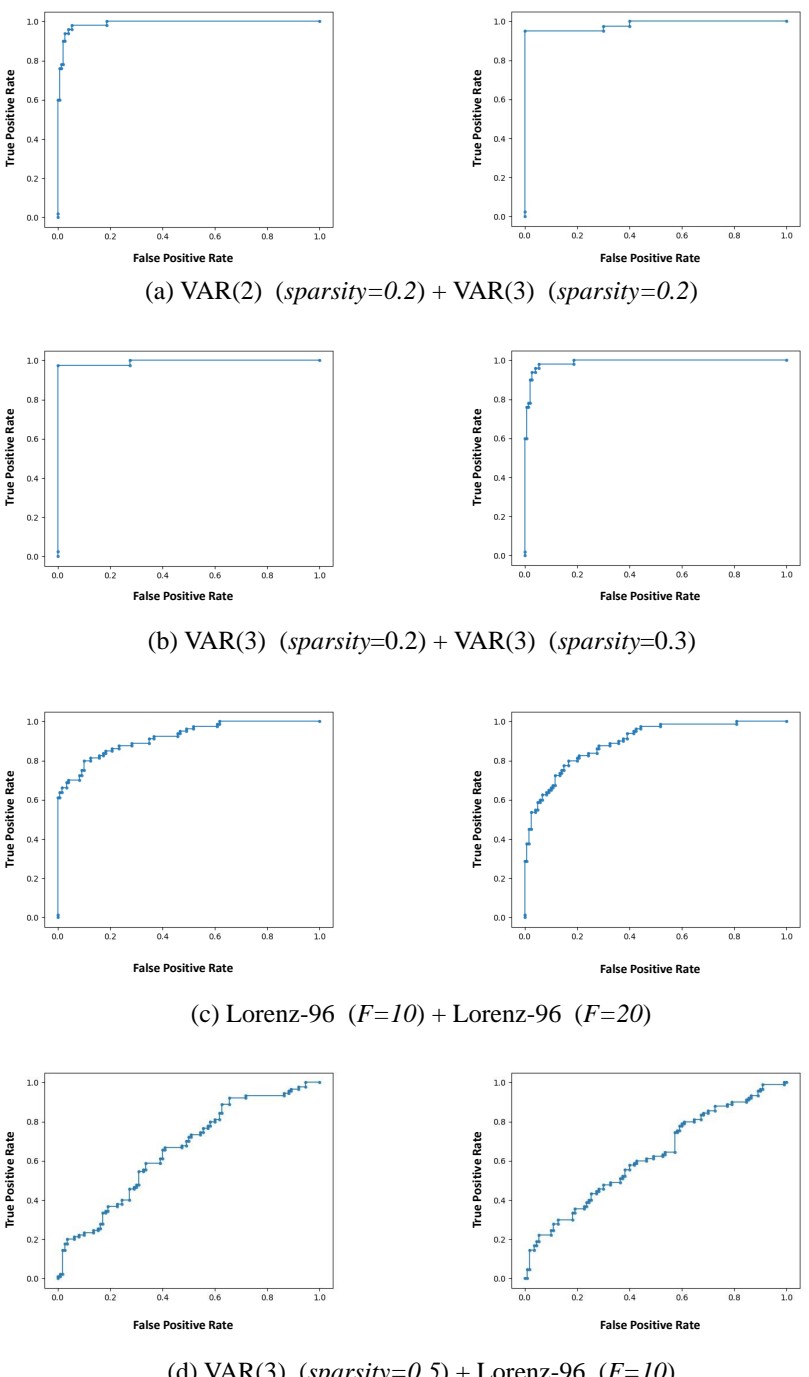

(a) VAR(2) (*sparsity=0.2*) + VAR(3) (*sparsity=0.2*)

(b) VAR(3) (*sparsity=0.2*) + VAR(3) (*sparsity=0.3*)

(c) Lorenz-96 (*F=10*) + Lorenz-96 (*F=20*)

(d) VAR(3) (*sparsity=0.5*) + Lorenz-96 (*F=10*)

Figure 6: Comparison of ROC curves for four time-varying Granger causality inference scenarios in Section 4.4. (Left) Multi-level fine-tuning method on GC-Mixer. (Right) Splitting time series manually on GC-Mixer.

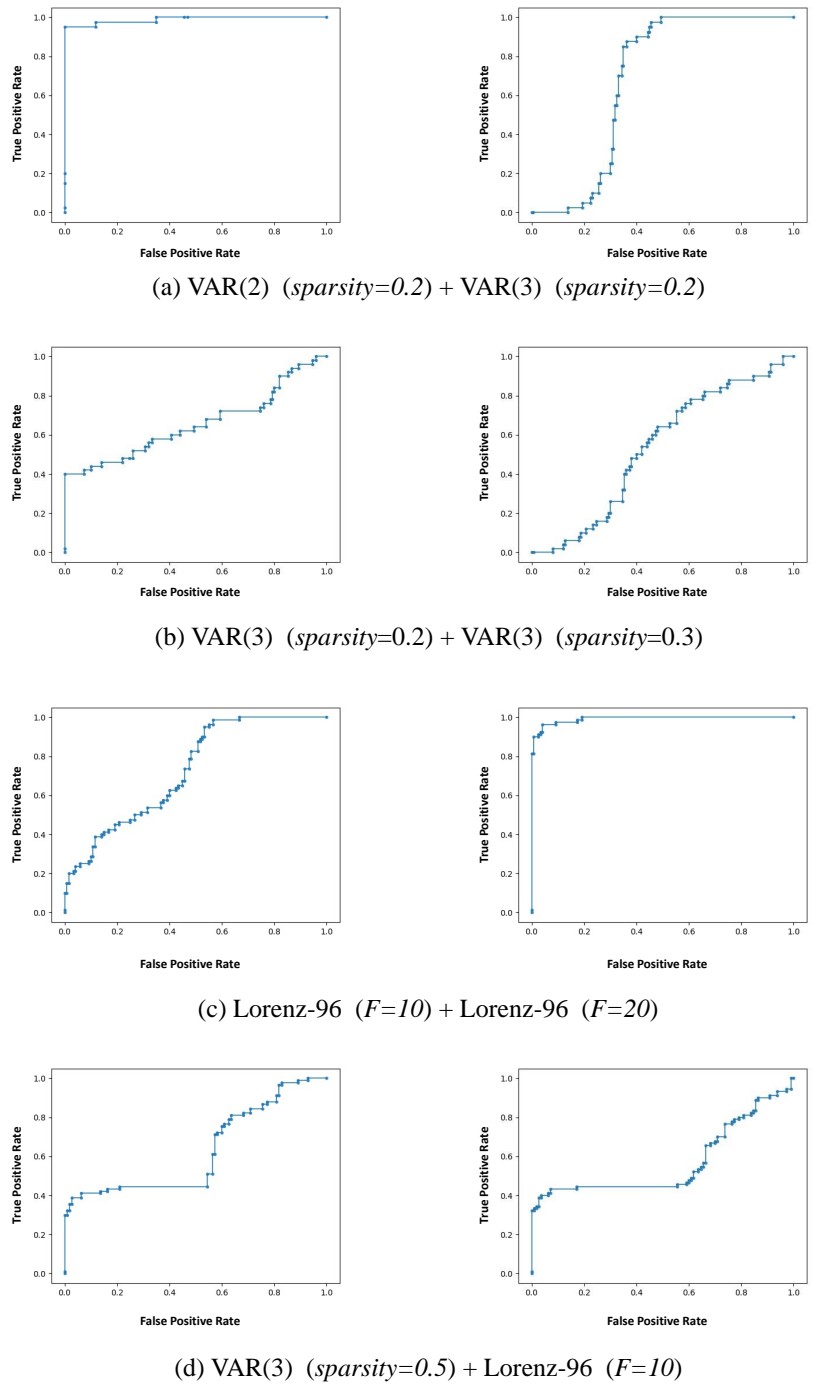

(a) VAR(2) (*sparsity=0.2*) + VAR(3) (*sparsity=0.2*)

(b) VAR(3) (*sparsity=0.2*) + VAR(3) (*sparsity=0.3*)

(c) Lorenz-96 (*F=10*) + Lorenz-96 (*F=20*)

(d) VAR(3) (*sparsity=0.5*) + Lorenz-96 (*F=10*)

Figure 7: Comparison of ROC curves for four time-varying Granger causality inference scenarios in Section 4.4. (Left) cMLP. (Right) cLSTM.

# D  APPENDIX FIGURE

The time-varying Granger causality inference of cMLP and cLSTM in Section 4.4 are shown in Figure 8.

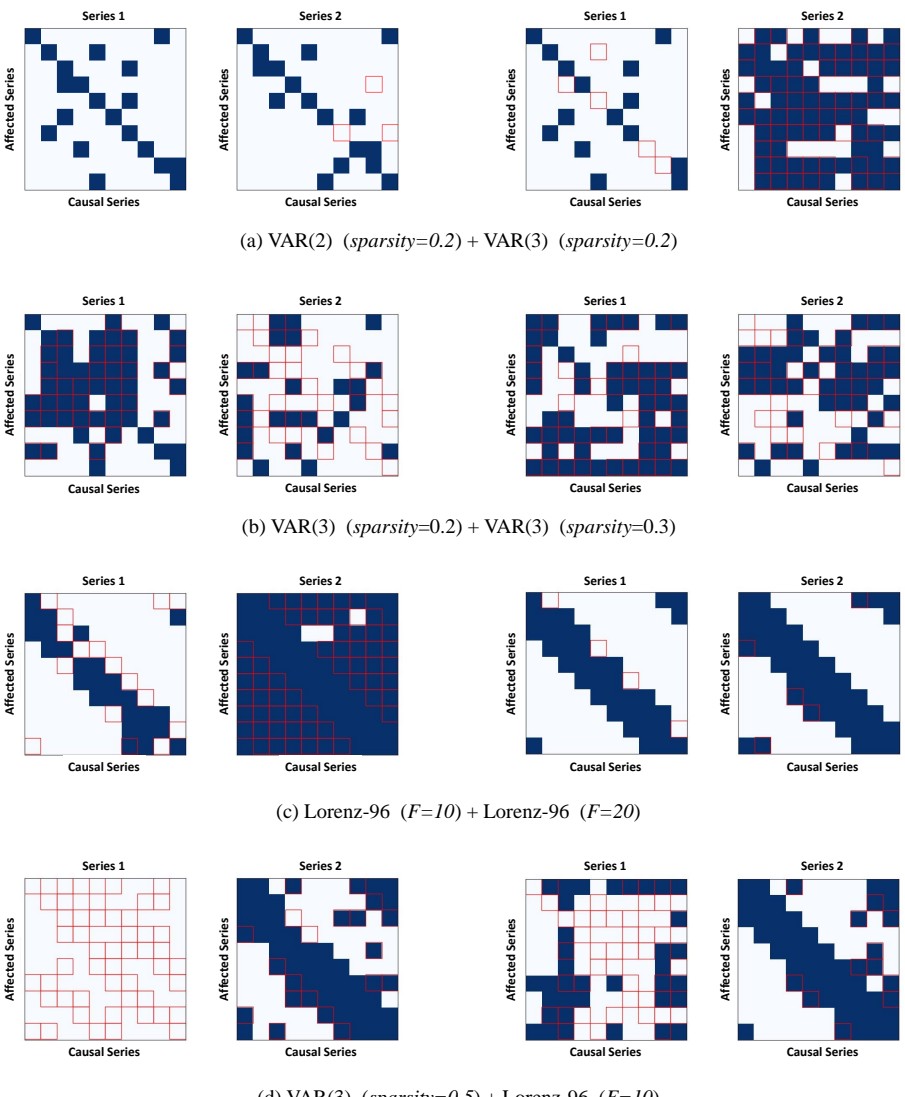

(a) VAR(2) (*sparsity=0.2*) + VAR(3) (*sparsity=0.2*)

(b) VAR(3) (*sparsity=0.2*) + VAR(3) (*sparsity=0.3*)

(c) Lorenz-96 (*F=10*) + Lorenz-96 (*F=20*)

(d) VAR(3) (*sparsity=0.5*) + Lorenz-96 (*F=10*)

Figure 8: Time-varying Granger causality inference. (Left) The two columns are inferred by cMLP. (Right) The two columns are inferred by cLSTM. The blue blocks indicate that Granger causality relationship exists between two time series. The white blocks indicate no Granger causality relationship between two time series. The blocks surrounded by the red line are the false Granger causality inferences.

# E  ABLATION RESULTS

In the following part, we will carry out ablation results to better understand our model. We changed the number of Mixer Blocks in our model, and the results are shown in Table 9 and Table 10.

Table 9: GC-Mixer, VAR (3), $T = 1000$, $p = 10$, $\lambda = 5e - 07$, optimizer: Adam

| Number of Block | AUROC | | | |
|---|---|---|---|---|
| | $sparsity = 0.2$ | $sparsity = 0.3$ | $sparsity = 0.4$ | $sparsity = 0.5$ |
| 1 | **1** | **1** | **1** | 0.994 |
| 2 | **1** | **1** | **1** | 0.984 |
| 3 | **1** | **1** | **1** | **0.999** |
| 4 | **1** | **1** | **1** | 0.989 |
| 5 | **1** | **1** | **1** | 0.988 |

Table 10: GC-Mixer, Lorenz-96, $T = 1000$, $p = 10$, $\lambda = 0.025$, optimizer: Adam

| Number of Block | AUROC | | | |
|---|---|---|---|---|
| | $F = 10$ | $F = 20$ | $F = 30$ | $F = 40$ |
| 1 | **0.943** | **0.921** | 0.801 | 0.731 |
| 2 | 0.687 | 0.829 | 0.835 | 0.735 |
| 3 | 0.765 | 0.830 | 0.846 | 0.817 |
| 4 | 0.643 | 0.739 | 0.824 | 0.801 |
| 5 | 0.705 | 0.819 | **0.869** | **0.863** |

