# OpenReview forum: "GC-Mixer: A Novel Architecture for Time-varying Granger Causality Inference"
_ICLR.cc/2024/Conference — Submitted to ICLR 2024_

### Official Review · Reviewer_NC5z · 2023-10-31

**Soundness:** 2 fair
**Presentation:** 2 fair
**Contribution:** 2 fair
**Rating:** 5
**Confidence:** 4

**Summary:**

In this paper, the authors investigate the problem of the Granger causal structure discovery. Considering that the existing method can hardly address the time-varying Granger-Causality inference, the authors proposed the GC-Mixer, which contains a mixer Block and a causality inference block. The authors further devise the multi-level fine-tuning method. The authors evaluate the proposed method on the VAR and Lorenz-96 datasets.

**Strengths:**

The authors address the problem of the time-varying Granger Causality inference.

**Weaknesses:**

1.	There are several methods that are proposed to solve the Granger Causality, for example [1],[2],[3][4][5]. Moreover, [2] is similar to the proposed method. Hence, it is suggested that the authors should discuss the difference between the proposed method and these methods and consider them as baseline.
2.	The authors propose multi-level fine-tuning to address the time-varying causal structure, but the motivation and intuition are not clear. Moreover, it is suggested that the authors should provide the complexity analysis for the multi-level fine-tuning method.
3.	I am also curious if the proposed method can address the causal structures that do not exist in the training set (OOD causal structure).



[1] CUTS: Neural Causal Discovery from Irregular Time-Series Data
[2] Interpretable Models for Granger Causality Using Self-explaining Neural Networks
[3] GRANGER CAUSAL INFERENCE ON DAGS IDENTIFIES GENOMIC LOCI REGULATING TRANSCRIPTION
[4] Causal Discovery from Temporal Data
[5] Neural Time-Invariant Causal Discovery from Time Series Data

**Questions:**

N.A.

---

> ### Author Response · Authors · 2023-11-17
>
> We thank the reviewer for the valuable comments. We address all the points as follows.
>
> Q1. There are several methods that are proposed to solve the Granger Causality, for example [1],[2],[3][4][5]. Moreover, [2] is similar to the proposed method. It is suggested that the authors should discuss the difference between the proposed method and these methods and consider them as baseline.
>
> We have supplemented the comparison result of the study [2] in our paper, as shown in table 1, 2, 3. While the comparison results of other studies are still being tested, we are temporarily unable to provide test results.
>
>
> Q2. The authors propose multi-level fine-tuning to address the time-varying causal structure, but the motivation and intuition are not clear. Moreover, it is suggested that the authors should provide the complexity analysis for the multi-level fine-tuning method.
>
> We initially manually divided the time series into multiple sequences and fed them into the model to infer Granger causality. However, we encountered issues with overfitting, and did not know how many segments would be best for splitting. Therefore, our motivation is to propose an algorithm that can automatically split the time series and make the model less prone to overfitting than before. The complexity analysis is updated and shown in Section 3.2.
>
> Q3: I am also curious if the proposed method can address the causal structures that do not exist in the training set (OOD causal structure).
>
> We have not conducted tests on the training set with an out-of-distribution causal structure and plan to incorporate such a dataset into our future experiments.

---

### Official Review · Reviewer_mLYv · 2023-10-31

**Soundness:** 3 good
**Presentation:** 2 fair
**Contribution:** 3 good
**Rating:** 6
**Confidence:** 2

**Summary:**

The authors propose a model for time-invariant Granger causality inference which exhibits consistent performance across various time series using the same hyperparameters. The model can also be extended to infer the time-varying Granger causality within a multi-level fine-tuning framework.

**Strengths:**

1. The literature review is comprehensive and clear.
2. The authors conduct extensive experiments on simulated datasets with different configurations.
3. The authors proposed an algorithm that has been claimed as the first algorithm to utilize an MLP Mixer-based architecture for inferring Granger causality in time series.

**Weaknesses:**

1. Given that the proposed algorithm involves significantly more parameters than the baselines, it would be helpful to include a comparison of the number of parameters required in the experiment section, along with the corresponding training times.
2. Some details about the loss function are not clear, which has been stated in the Questions section.
3. Some details about the experiments are not clear, which has been stated in the Questions section.

**Questions:**

1. In the equation 12 loss function, the right-hand side of the equation contains the time index $t$ and time series index $i$, and there is no summation regarding $t$ and $i$. Does this mean the loss function is computed for each time series $i$ at a specific time $t$? Personally, I do not think the loss function is linked to each $t$, but equation 12 seems to suggest otherwise.
2. What is the value of the threshold $\epsilon$ in equation 13 in the experiment, and how to determine it? Could you help me locate the place if you have stated it already in the paper?
3. It is advisable to cite the related work in the first paragraph in section 2.3 regarding the existing approach?
4. Can you clarify how you compute the True Positive Rate (TPR) and False Positive Rate (FPR)? Do these metrics calculate from the F-norm of $W_{j,k}^n$? In other words, do they not only assess whether time series $j$ Granger-causes time series $i$ but also consider whether the lag is accurate? For instance, if the true lags are $1,2,3$, would lag $4$ from time series $j$ to $i$ be regarded as a False Positive?
5. Given that the loss function is computed for each time series $i$, does this mean the whole algorithm will run $p$ times for $p$-variate time series?
6. Table 3 shows that the cMLP algorithm outperforms the proposed algorithm. Since no other nonlinear experiment has been provided, does this imply that the proposed algorithm performs worse than the baselines in the nonlinear cases? Personally, I recommend conducting further nonlinear experiments to thoroughly examine the performance of the proposed algorithm in nonlinear cases.
7. Can additional nonlinear experiments be conducted in section 3.3 on automatic lag selection, given that the Lorenz-96 dataset does not involve time lag? Since the proposed method did not yield the best performance in the Lorenz-96 dataset, might this also be the case in the time lag selection stage? Additional results would provide valuable insights.
8. Can you offer more details regarding the procedure for generating the time series in section 3.4 on time-varying Granger causality inference? If, in the four scenarios, two sets of time series with different configurations are merged together with equal lengths, would this potentially favor the proposed algorithm? Considering that the multilevel fine-tuning algorithm separates input time series into $2^{i-1}$ segments, such a configuration might be advantageous. Furthermore, could you provide the information about the value of $i$ when the algorithm stops in the time-varying experiment?
9. Still in section 3.4, how to do manual splitting? Does it utilize additional information about the true time series?
10. Could you briefly explain why the algorithms mentioned in section A.2 have not been applied as baselines, though they ask to select the time lag manually?
11. It is advisable to enhance the visualization by incorporating axis titles and subtitles.

---

> ### Author Response · Authors · 2023-11-17
>
> We thank the reviewer for the valuable comments and correcting our mistakes. We address all the points as follows.
>
>  Q1. Does this mean the loss function is computed for each time series $i$ at a specific time $t$?
>
> The loss function is independent of $t$, and we have corrected this mistake in the Equation 12.
>
> Q2. how to determine threshold in equation 13 in the experiment?
>
> Firstly, we generate the AUROC using one hyperparameter lambda and sweep threshold epsilon. Then, when the AUROC curve takes the optimal operating point (the point closest to the upper-left corner), the corresponding epsilon are taken to infer the Granger causality.
>
> Q3: It is advisable to cite the related work in the first paragraph in section 2.3 regarding the existing approach?
>
> We have added references in the corresponding section.
>
> Q4: Can you clarify how you compute the True Positive Rate (TPR) and False Positive Rate (FPR)?  Do these metrics calculate from the F-norm of W?  Do they not only assess whether time series $j$ Granger-causes time series $i$ but also consider whether the lag is accurate?
>
> Yes, the TPR and FPR are calculated from the F-norm of W and did not consider whether the lag is accurate. This is why we still need to compare the performance of cMLP and GC-Mixer on automatic lag selection in Section 4.3.
>
> Q5. Does this mean the whole algorithm will run $p$ times for $p$-variate time series?
>
> Yes, the whole algorithm will run $p$ time for $p$-dimensional time series.
>
> Q6. Foes this imply that the proposed algorithm performs worse than the baselines in the nonlinear cases? Personally, I recommend conducting further nonlinear experiments to thoroughly examine the performance of the proposed algorithm in nonlinear cases.
>
> We have begun to evaluate performance on more nonlinear datasets, including Dream3 and Lotka–Volterra. Due to time limitations, we are unable to immediately provide specific results.
>
> Q7. Can additional nonlinear experiments be conducted in section 3.3 on automatic lag selection, given that the Lorenz-96 dataset does not involve time lag?
>
> We considered adding more nonlinear datasets to test the performance of automatic time lag selection. Unfortunately, neither the datasets provided in the existing literatures, including Lorenz-96, Dream-3, FMRI Bold, and Lotka–Volterra, have no time lag. We are actively looking for nonlinear datasets including time lags, but for now, we cannot test the performance of automatic lag selection on nonlinear datasets.
>
> Q8. Can you offer more details regarding the procedure for generating the time series in section 3.4 on time-varying Granger causality inference? Could you provide the information about the value of $i$ when the algorithm stops in the time-varying experiment?
>
> Such a configuration will indeed be beneficial to our proposed algorithm. However, the comparison of algorithms is under the same configuration, so it can reflect the performance of the proposed algorithm. Moreover, four values of $i$ when the algorithm stops for four scenarios are 3.
>
> Q9. Still in section 3.4, how to do manual splitting? Does it utilize additional information about the true time series?
>
> We manual split the sequence into two equal-length time series and use the same lambda for training. In addition, the time-varying data we currently test is still generated by simulation. Due to time constraints, our future work will conduct experiments on the real-time series.
>
> Q10. Could you briefly explain why the algorithms mentioned in section A.2 have not been applied as baselines, though they ask to select the time lag manually?
>
> Our research focuses on deep learning models, which differ from traditional time-varying Granger causality models. Therefore, we have chosen not to conduct a direct comparison with traditional models in this paper, but we have planned to conduct an exhaustive comparison of our proposed algorithm with the related algorithms mentioned in Section A.2 in future research.
>
> Q11: It is advisable to enhance the visualization by incorporating axis titles and subtitles.
>
> We have updated the axis titles in Figure 5, 6, 7, 8.

---

> ### Comment · Reviewer_mLYv · 2023-11-22
>
> Thank you for all the clarifications. While most of my inquiries have been resolved, I still have a couple of remaining questions:
>
> 1. Can you briefly address the first point outlined in the Weakness section regarding the number of parameters needed and the training time?
> 2. Regarding the answer for Q6, could you provide an approximate estimate of the time required to execute the algorithm on a dataset of a specific size?
> 3. Concerning the statement "The time complexity of the algorithm is O(log2n)" included in the main paper, does the variable "n" denote the number of time series subsequences? If so, should it be represented as "T − K + 1" instead of "n," considering that "n" denotes the subsequence index rather than the total number of subsequences? Additionally, is the complexity solely associated with "n" and not influenced by the dimension of the multivariate time series "p" or the level "i"?
> 4. Based on the answer provided for Q9, does it suggest that in the manual splitting scenario, "i"= 2, while in the automatic splitting scenario, "i" is algorithmically determined and turns out to be 3 across all four scenarios? Table 4 shows that for the proposed method, the outcome of "Multi-level fine-tuning (Automatic splitting)" doesn't significantly outperform the proposed method employing manual splitting. Considering the computational cost and the parameter requirements associated with the multi-level approach, does this indicate that manual splitting might be preferable from a practical standpoint?

---

> ### Author Response · Authors · 2023-11-22
>
> We thank the reviewer for the valuable comments and correcting our mistakes. We address all the points as follows.
>
> Q1: Does the "first point outlined in the Weakness section" refer to the sentence that "GC-Mixer has more parameters than cMLP and cLSTM, leading to more prone to overfit"? If so, when the number of the Mixer Block in GC-Mixer is 1, the total number of parameters of GC-Mixer is 485520. The total number of parameters of cMLP is 52010, and cLSTM is 449010. Our model contains more parameters than cMLP and cLSTM.
>
> Q2: Perhaps the time comparison of running GC-Mixer and cMLP on the same computer could answer this question. For VAR (3) (sparsity =0.2), the time consumption of GC-Mixer (optimizer: Adam) is about 150s; cMLP (optimizer: ISTA) is over 1400s, cMLP (optimizer: Adam) is only 5s. For Lorenz-96 (F=10), the time consumption of GC-Mixer (optimizer: Adam) is about 700s, and cMLP (optimizer: ISTA) is about 70s (early stop); cMLP (optimizer: Adam) is about 5s. (By using ISTA, not Adam optimizer, cMLP can achieve the best performance AUROC.)
>
>
> Q3: Yes, we have made a mistake. The time complexity is related to the number of subsequence indexes $(T-K+1)$, the dimension p, and the level i, that is, $O((T-K+1) \times p \times (2^i-1))$. In addition, in our code, $T-K+1$ subsequences are calculated in parallel so that the actual complexity will be less than $O((T-K+1) \times p \times (2^i-1))$.
>
> Q4: Yes, we split the time series into two sequences for manual splitting. For automatic splitting, the algorithm splits the time series into four sequences (i=3).
> For the second question, in our experiments, we adopted the optimal splitting number for the manual splitting: to directly split the sequence into two for calculation. However, for a practical scenario, we may not know the optimal number of splitting, so using an algorithm to split automatically may be a better option.

---

> > ### Comment · Reviewer_mLYv · 2023-11-22
> >
> > Thank you for answering those follow-up questions.
> >
> > The fourth question revolves around the observation that although an optimal number of splits exists, Table 4 indicates no substantial difference between manual and automatic splitting. In such a scenario, why do people choose the latter option despite requiring more parameters and longer training time?
> >
> > For a clearer demonstration of the importance of automatic splitting, would it be more effective to conduct experiments using an optimal number of splits greater than 2? This could highlight how automatically determined splits align closely with the optimal choice and yield significantly better results compared to manual splitting, especially when using a smaller or incorrect number of splits in the manual splitting, e.g,, 2.

---

> > > ### Author Response · Authors · 2023-11-23
> > >
> > > We thank the reviewer for the valuable comment.
> > >
> > > Following your suggestion, we manually split the sequence into four segments. Due to time constraints, we only obtained the results of scenario 3 (Lorenz 96, F=10 + F=20). The average AUROC of 5 experiments is 0.75, lower than the 0.92 of multi-level fine-tuning. Moreover, we are expanding our experiment by using a smaller or incorrect number of splits. The comparison results will be updated in our paper later.

---

> > > > ### Comment · Reviewer_mLYv · 2023-11-23
> > > >
> > > > Thanks for carrying out additional experiments. I don't have any more questions.

---

### Official Review · Reviewer_KhHm · 2023-11-01

**Soundness:** 3 good
**Presentation:** 3 good
**Contribution:** 2 fair
**Rating:** 3
**Confidence:** 4

**Summary:**

This work considers the topic of Granger causality discovery in multivariate time series and focuses on how to do so using deep learning. The main contribution is a new method called GC-Mixer that leverages an alternative network architecture, and which shows promising results in GC inference with synthetic datasets where the ground truth is known. In addition, the authors propose a method for automatically splitting a long time series to discover time-varying GC dynamics.

**Strengths:**

Granger causality discovery is a difficult problem which is not fully solved by current methods, including those using deep learning. Capturing nonlinear dynamics during GC inference is challenging, and neural networks can help alleviate this problem. However, current approaches can be cumbersome to train and offer performance that is far from perfect, even on these relatively simple datasets. It is therefore worthwhile to pursue alternative approaches like this one, which leverage alternative network architectures. On top of that, it's important to explore solutions for detecting time-varying Granger causal relationships.

**Weaknesses:**

Several questions and concerns:

- It was difficult to follow the description of the mixer architecture, which is one of the main contributions of this work. If I understand correctly, it seems like the second projection in the mixer block, shown in eq. 8, actually mixes information between all the time series. If that's true, wouldn't it mean that every prediction depends on every time series? That should make it difficult to identify which predictors are important, because every forecast automatically depends on every input time series.

- Again, it was hard to follow the description of the network architecture, but it seems like the predictions are ultimately based on an element-wise multiplication between the inputs and the output of the mixer block (this is then fed to a MLP). Would it be fair to interpret these as attention weights? It might be a helpful analogy, because similar notions of soft attention have been used in transparent deep learning.

- I'll temporarily assume, following my question above, that the $W^{(n)}$ values can be viewed as attention weights. When we pass the attention-weighted inputs $M$ into the MLPs $g_i$, do we use separate attention weights for each $g_i$? Otherwise, it would seem that we're forced to make one set of selections for all predictions, whereas we should instead select the relevant inputs for the prediction corresponding to each output series. Either I'm missing something in the notation, or this seems like a restrictive choice.

- It seems inconvenient that the weights $W^{(n)}$ determine the Granger causality relationships, but that they vary for every time point. Compared to the cMLP/cLSTM, it means that an input can be deemed non-causal only if it has small weights for all time points. Is that correct?

- The authors claim that they cannot make the $W^{(n)}$ weights exactly equal to zero, even with the group lasso penalty. This seems correct, because the weights are the output of the mixer block. However, it is untrue that the cMLP/cLSTM share this issue: they regularize parameters of the network, and the parameters can reach zero exactly due to optimization with proximal updates.

- The authors state that they applied the hierarchical penalty to all models tested here, including GC-Mixer, cMLP and cLSTM. However, the cLSTM only has an explicit dependence on one past timepoint, so it's not possible to apply the hierarchical penalty. Indeed, Tank et al discussed that penalty only in the context of the cMLP. Can the authors explain what they mean about using the hierarchical penalty with the cLSTM, because this sounds like a mistake.

- The methods for splitting the time series to discover time-varying dynamics seems reasonable. However, the experiment that tests it seems quite simplistic, and I wonder if the authors could design an experiment that is either more challenging or more realistic. Also, I'm not certain about this, but it seems like the algorithm has no specific relationship with GC-Mixer: it is perhaps unfair to only use it with GC-Mixer in Table 4 and not apply it with cMLP or cLSTM?

- The results with VAR data are encouraging, but this is not the type of data where GC-Mixer should be most valuable. Indeed, we would expect that traditional linear methods would perform far better with this data. On the Lorenz dataset that's actually nonlinear, GC-Mixer underperforms both cMLP and cLSTM.

- The results in Figure 4 look like they did not involve tuning the penalty strength for cMLP. Can the authors describe what they did here and whether it's providing a fair comparison?

- The final architecture is quite complicated, and I wonder if the authors performed any ablations to understand what aspects are important for it to work. For example, could they try with different numbers of blocks? Or removing batch normalization? Currently, it is hard to understand why this type of autoregressive model should enable better GC discovery than a MLP or LSTM - it's a different parameterization, but not obviously better. (And as mentioned above, the empirical results are not convincing on their own.)

- Could the authors provide more details about how they generated AUROC curves? For example, did they keep epsilon fixed and train with different lambda values? Or did they train with one lambda value and sweep epsilon? It would be important to know whether there are any differences compared to previous methods.

- For the title of Section 2.1.2, the "non-autoregressive" model looks like it actually is autoregressive, in that it predicts the future using the past. Perhaps what the authors meant to say is that it's nonlinear? This should be corrected.

- Typo in Section 2.2.3: GULE -> GELU

**Questions:**

Several questions are mentioned in the weaknesses section above.

---

> ### Author Response · Authors · 2023-11-17
>
> We thank the reviewer for the valuable comments and correcting our mistakes. We address all the points as follows.
>
> Q1. Wouldn't it mean that every prediction depends on every time series?
>
> Yes, every prediction depends on every time series. According to Tank et al., 2021, they proposed component-wise MLP and tackled the challenge that each $X_{ti}$ may depend on different past lags of the other series. Therefore, our mixer architecture will not make it difficult to identify which predictors are important.
>
> Q2. The predictions are ultimately based on an element-wise multiplication between the inputs and the output of the mixer block (this is then fed to a MLP). Would it be fair to interpret these as attention weights?
>
> First, we correct the description of W to the causal matrix in our paper, which can better describe W than the previous weight matrix (attention weights). Furthermore, this operation does have some similarities with the soft attention mechanism. Our model adjusts attention to the input time sequences by learning weights for better prediction. The learned weight is the impact of a time lag k of a time series j on the prediction of $X_{ti}$ (due to the matrix being element-wise multiplicated with input time series). Therefore, the matrix W can be used to infer Granger causality after the hierarchical group lasso penalty.
>
> Q3. Do we use separate attention weights for each $g_{i}$?
>
> Yes, we use separate $W$ for each $g_{i}$. We have corrected the equation 10, 11, 12, 13 and give a more detailed the description about W in Section 3.1.3.
>
> Q4.  Compared to the cMLP/cLSTM, it means that an input can be deemed non-causal only if it has small weights for all time points. Is that correct?
>
> Yes, according to Equation 13, time series $i$ not Granger-cause to series $j$ only if it has small weights for all time points.
>
> Q5. It is untrue that the cMLP/cLSTM share this issue: they regularize parameters of the network, and the parameters can reach zero exactly due to optimization with proximal updates.
>
> Whether the proximal update optimization can penalize the parameter to zero is relative to the lambda value. Here, we give two test cases (cMLP; lambda: 5; Lorenz-96; F:40; optimizer: ISTA), the AUROC we tested is 0.926. The proximal update optimization did not penalize any parameters of the weight matrix in cMLP to zero. However, for another test case (cMLP; lambda: 20; Lorenz-96; F:40; optimizer: ISTA), the AUROC we tested is 0.932. In this case, the proximal update optimization indeed penalizes most parameters of the weight matrix in cMLP to zero. Therefore, the proximal update optimization can regularize parameters to zero, but experimental results show that it does not work in all cases.
>
> Q6: Can the authors explain what they mean about using the hierarchical penalty with the cLSTM, because this sounds like a mistake.
>
> Yes, it is a mistake, cLSTM doesn’t use hierarchical penalty, we have corrected this mistake in Section 3.1.3.
>
> Q7.  It is perhaps unfair to only use it with GC-Mixer in Table 4 and not apply it with cMLP or cLSTM?
>
> Following your advice, we have applied the proposed multi-level fine-tuning algorithm on cMLP and cLSTM. The results are updated in Table 4.
>
> Q8. The results with VAR data are encouraging, but on the Lorenz dataset, GC-Mixer underperforms both cMLP and cLSTM.
>
> In our initial tests, we found that cMLP and cLSTM performed excellent on the Lorenz-96 dataset, and their AUROC could be close to 1 in many cases. Unfortunately, these two models performed unsatisfactorily on the VAR dataset. Therefore, our motivation is to propose a model that performs well on both datasets, allowing our model to be adapted for both time-invariant and time-varying scenarios.
>
> Q9. The results in Figure 4 look like they did not involve tuning the penalty strength for cMLP. Can the authors describe what they did here and whether it's providing a fair comparison?
>
> We selected the best-performing penalty strength for cMLP and GC-Mixer on sparsity = 0.2. Then, we kept the penalty strength unchanged for both cMLP and GC-Mixer and conduct the experiment on sparsity = 0.3. Therefore, it was a fair comparison.
>
> Q10.  Could they try with different numbers of blocks? Or removing batch normalization?
>
> Currently, we are engaged in ablation experiments to assess the results when altering the number of blocks or removing batch normalization from the model and find what aspects are important for it to work.
>
> Q11. Could the authors provide more details about how they generated AUROC curves?
>
> The AUROC is generated with one lambda value and sweep epsilon.
>
> Q12. Perhaps what the authors meant to say is that it's nonlinear?
>
> We have corrected the title of Section 3.2 “non-autoregressive” into “nonlinear autoregressive”.
>
> Q13. Typo in Section 2.2.3: GULE -> GELU
>
> We have corrected this mistake in Section 3.1.3.

---

> > ### Comment · Reviewer_KhHm · 2023-11-20
> > **Response**
> >
> > Thanks to the authors for their edits to the paper. Some thoughts on the revisions and clarifications:
> >
> > Q1. It's worth acknowledging that this is strange design choice. The point of sparsity in Tank et al is to fit networks that eliminate dependence on certain inputs. Here, dependence is eliminated via sparsity in $W$, but $W$ itself still depends on all the inputs. It seems possible for a certain input to be truly Granger causal, yet receive no weight via $W$ because its role is to determine which other inputs should be used in the forecast (i.e., it acts like a gating variable). This is kind of a hypothetical concern that may or may not show up with real datasets, so including more realistic ones would be helpful.
> >
> > Q5. The difference is still notable: choosing a high $\lambda$ value can ensure that the cMLP/cLSTM eliminates certain input dependencies. For this approach, because we're attempting to sparsify predictions rather than weights, I'm not sure we can guarantee sparsity regardless of the $\lambda$ value.
> >
> > Q8. It makes this method significantly less interesting that it underperforms on nonlinear data. Also, I wonder if it's especially well matched for VAR data because $W$ can basically be a constant prediction for all timepoints, $g$ can behave linearly, and we'll recover the exact VAR model. I didn't realize this before, but now that I do I think it's important to expand the experiments before publication.
> >
> > Q10. I think these ablation results would be important to include before publication.
> >
> > Q11. I'm pretty sure this differs from how Tank et al generated results: they effectively set $\epsilon = 0$ and swept $\lambda$. How do the authors set $\lambda$ before sweeping $\epsilon$? This seems like an important hyperparameter choice.

---

> > > ### Author Response · Authors · 2023-11-22
> > >
> > > Q1. Yes, more realistic experiments would be helpful for this concern. We are testing our model on more realistic datasets, including Dream-3 and FMRI Bold following your suggestions.
> > >
> > > Q8. Following your suggestions, we are expanding the experiments on the VAR dataset.
> > >
> > > Q10. We have conducted the ablation experiments by changing the number of Block. The results are shown in Table 9 and Table 10 in the Appendix E.
> > >
> > > Q11. Yes, the generation of AUROC mainly refers to study [1], which differs from the Tank et al. The setting of the lambda: We traverse the lambda, and for each lambda, we sweep the epsilon to get the corresponding AUROC. Finally, the best-performance AUROC and the corresponding lambda are selected.
> > >
> > > [1] Ricards Marcinkevics, Julia E. Vogt: Interpretable Models for Granger Causality Using Self-explaining Neural Networks. ICLR 2021

---

### Official Review · Reviewer_1qK9 · 2023-11-09

**Soundness:** 2 fair
**Presentation:** 2 fair
**Contribution:** 2 fair
**Rating:** 3
**Confidence:** 4

**Summary:**

The paper studies the Granger Causality inference with the proposed GC-Mixer model. The research topic is interesting, but the paper seems to lack unique motivation and contribution. Also, the theoretical contribution and empirical analysis of the paper are inadequate.

**Strengths:**

- The research topic is interesting, using deep learning tools for effectively capturing the non-linear Granger Causality.

- The paper is somehow easy to follow.

**Weaknesses:**

- The novelty and the motivation of the paper are not clear.

- Many state-of-the-art Granger Causality studies like [1,2,3] are missed or not fully discussed and compared in the paper.

- The theoretical contribution seems inadequate.

- The experiments seem weak, to some extent.

- The organization of the paper is busy and can be improved, the authors may want to split Section 2.

[1] Saurabh Khanna, Vincent Y. F. Tan: Economy Statistical Recurrent Units For Inferring Nonlinear Granger Causality. ICLR 2020

[2] Ricards Marcinkevics, Julia E. Vogt: Interpretable Models for Granger Causality Using Self-explaining Neural Networks. ICLR 2021

[3] Wenbo Gong, Joel Jennings, Cheng Zhang, Nick Pawlowski: Rhino: Deep Causal Temporal Relationship Learning with History-dependent Noise. ICLR 2023

**Questions:**

What is the unique motivation and novel contribution of the paper that set it apart from previous studies?

---

> ### Author Response · Authors · 2023-11-17
>
> We thank the reviewer for the valuable comments. We address all the points as follows.
>
> Q1: The novelty and the motivation of the paper are not clear.
>
> In our preliminary evaluations, cMLP and cLSTM demonstrated outstanding performance on the Lorenz-96 dataset, with AUROC values often approaching 1. Unfortunately, these two models performed unsatisfactorily on the VAR dataset. Therefore, our motivation is to propose a model that performs well on both datasets, which can adapt for both time-invariant and time-varying scenarios. The novelty of this paper: We introduce a new approach to extract time-invariance Granger causality from the output of Mixer Block in our model, which is different from other models. Furthermore, we propose a new algorithm that enables the model to infer time-varying Granger causality.
>
> Q2: Many state-of-the-art Granger Causality studies like [1,2,3] are missed or not fully discussed and compared in the paper.
>
> We have supplemented the comparison results of the study [2], as shown in Tables 1, 2, and 3. While the comparison of studies [1] and [3] are still being tested, we are temporarily unable to provide test results.
>
>
> Q3: The experiments seem weak, to some extent.
>
> We have begun to evaluate performance on more datasets, including Dream3 and Lotka–Volterra. Due to time limitations, we are unable to immediately provide specific results.
>
> Q4: The organization of the paper is busy and can be improved, the authors may want to split Section 2.
>
> We have split Section 2 into two sections following your suggestion.
>
> [1] Saurabh Khanna, Vincent Y. F. Tan: Economy Statistical Recurrent Units For Inferring Nonlinear Granger Causality. ICLR 2020
>
> [2] Ricards Marcinkevics, Julia E. Vogt: Interpretable Models for Granger Causality Using Self-explaining Neural Networks. ICLR 2021
>
> [3] Wenbo Gong, Joel Jennings, Cheng Zhang, Nick Pawlowski: Rhino: Deep Causal Temporal Relationship Learning with History-dependent Noise. ICLR 2023

---

### Meta-Review · Area_Chair_arhp · 2023-12-09

**Metareview:**

The paper introduces the GC-Mixer model for Granger causality inference in multivariate time series, aiming to capture nonlinear causality using deep learning tools. The reviewers have expressed mixed opinions on the paper's merits and shortcomings.

The strengths lie in 1) The research topic of Granger causality inference in multivariate time series using deep learning is acknowledged as interesting and challenging. 2) The paper is considered somewhat easy to follow, and the clarity of writing and presentation is recognized.

The weaknesses lie in 1) The paper lacks clear motivation and a distinct contribution, and the novelty of the proposed approach is not well-established. 2) Several state-of-the-art Granger causality studies are missed or not adequately discussed and compared.
3) The theoretical contribution is deemed inadequate, and the experiments are criticized for being weak to some extent.
4) The organization of the paper is considered busy and could be improved, with suggestions to split Section 2.
5) Specific questions and concerns are raised regarding the model architecture, attention weights interpretation, penalty application, and experimental design.

The weaknesses, especially in motivation, clarity, and experimental design, are highlighted consistently. The authors' responses indicate attempts to improve the paper, but the overall recommendation leans towards rejection.

**Justification For Why Not Higher Score:**

The weaknesses, especially in motivation, clarity, and experimental design, are highlighted consistently. The authors' responses indicate attempts to improve the paper, but the overall recommendation leans towards rejection.

**Justification For Why Not Lower Score:**

N/A

---

### Decision · Program_Chairs · 2024-01-16

Reject